# The Feasibility and Safety of Temporary Transcatheter Balloon Occlusion of Bilateral Internal Iliac Arteries during Cesarean Section in a Hybrid Operating Room for Placenta Previa with a High Risk of Massive Hemorrhage

**DOI:** 10.3390/jcm11082160

**Published:** 2022-04-12

**Authors:** Jin-Gon Bae, Young Hwan Kim, Jin Young Kim, Mu Sook Lee

**Affiliations:** 1Department of Obstetrics and Gynecology, Keimyung University Dongsan Hospital, School of Medicine, Keimyung University, Daegu 42601, Korea; jgonmd@gmail.com; 2Department of Radiology, Keimyung University Dongsan Hospital, School of Medicine, Keimyung University, Daegu 42601, Korea; newhippo@naver.com (Y.H.K.); jinkim0411@naver.com (J.Y.K.)

**Keywords:** cesarean section, placenta abnormalities, temporary transcatheter balloon occlusion of the bilateral internal iliac arteries

## Abstract

This study aimed to evaluate the feasibility and safety of temporary transcatheter balloon occlusion of bilateral internal iliac arteries (TBOIIA) during cesarean section in a hybrid operating room (OR) for placenta previa (PP) with a high risk of massive hemorrhage. This retrospective study analyzed the medical records of 62 patients experiencing PP with a high risk of massive hemorrhage (mean age, 36.2 years; age range 28–45 years) who delivered a baby via planned cesarean section with TBOIIA in a hybrid OR between May 2019 and July 2021. Operation time, estimated blood loss (EBL), amount of intra- and postoperative blood transfusion, perioperative hemoglobin level, hospital stay after operation, balloon time, fluoroscopy time, radiation dose, rate of uterine artery embolization (UAE) and hysterectomy, and complication-related TBOIIA were assessed. The mean operation time was 122 min, and EBL was 1290 mL. Nine out of sixty-two patients (14.5%) received a blood transfusion. The mean hemoglobin levels before surgery, immediately after surgery and within 1 week after surgery were 11.3 g/dL, 10.4 g/dL and 9.2 g/dL, respectively. In terms of radiation dose, the mean dose area product (DAP) and cumulative air kerma were 0.017 Gy/cm^2^ and 0.023 Gy, respectively. Ten out of sixty-two patients (16.1%) underwent UAE postoperatively in the hybrid OR. One out of sixty-two patients had been diagnosed with placenta percreta with bladder invasion based on preoperative ultrasound, and thus underwent cesarean hysterectomy following TBOIIA and UAE. While intra-arterial balloon catheter placement for managing PP with a high risk of hemorrhage remains controversial, a planned cesarean section with TBOIIA in a hybrid OR is effective in eliminating the potential risk of intra-arterial balloon catheter displacement, thus reducing intraoperative blood loss, ensuring safe placental removal and conserving the uterus.

## 1. Introduction

Placenta previa (PP), where the placenta overlies the cervix, occurs in 4–5 out of 1000 pregnancies [1]. Although a trial of labor can be considered in carefully selected patients with PP, a cesarean section is recommended for patients with the placental edge within 2 cm from the cervical os [2], since all patients with PP have a risk of bleeding regardless of their placenta location or type [3]. In cases of PP with placenta accreta spectrum (PAS), the patients are at an increased risk of massive peripartum hemorrhage and hysterectomy [2,4].

Surgeries for placental abnormality are highly challenging compared to other cesarean operations [5]. In addition to the challenging operation itself, there are other complicators, including the risk of postoperative uncontrolled peripartum hemorrhage, consequent need for massive transfusion, and risk of complications, such as disseminated intravascular coagulation and multiorgan failure. For this reason, a multidisciplinary team approach involving experienced obstetricians, maternal–fetal medicine specialists, interventional radiologists, transfusion medicine specialists and critical care experts is crucial [6].

Several uterine devascularization techniques have been developed to control intraoperative and perioperative blood loss during surgery for placental abnormality, including surgical approaches involving ligation of the uterine artery or hypogastric artery, as well as endovascular approaches involving balloon occlusion or embolization [7]. Since Dubois et al. first reported a case in which intraoperative blood loss was reduced through balloon occlusion and embolization of the internal iliac arteries in a patient with placenta percreta [8], there have been studies on a variety of endovascular interventional modalities employed to control bleeding during abnormal placental implantation deliveries [9,10].

In addition, hybrid operating rooms (ORs) (an advanced surgical theater combining a general operating suite with interventional radiology departments) have become more useful [11], and they provide a safer environment than a conventional OR for the management of complex obstetric patients. Despite the advantages of a hybrid OR, few reports are available on its usefulness for patients experiencing PP with a high risk of hemorrhage [12].

The purpose of the present study was to evaluate the feasibility and safety of temporary transcatheter balloon occlusion of bilateral internal iliac arteries (TBOIIA) during cesarean section in a hybrid OR for PP with a high risk of massive hemorrhage.

## 2. Materials and Methods

### 2.1. Patients

Institutional review board approval was obtained for this study, and informed consent was waived because of its retrospective nature. Of 323 patients who underwent a cesarean section due to PP between May 2019 and July 2021, 81 patients with a risk of massive transfusion of ≥5 units of packed red blood cells during the cesarean section [13] underwent TBOIIA prior to the cesarean section (mean age, 36.1 years; age range, 24–46 years). Of these patients, 62 patients who had undergone planned cesarean section with TBOIIA in a hybrid OR (mean age, 36.2 years; age range, 28–45 years) were retrospectively analyzed (Figure 1). All patients and their families were informed of the risk of massive bleeding during surgery, consequent risk of complications and TBOIIA and signed a consent form prior to the surgery.

### 2.2. TBOIIA Procedures

In principle, TBOIIA (Figure 2a,b) is a one-step procedure that is performed prior to cesarean section in a hybrid OR with an image-guided interventional suite. First, local anesthesia was performed with lidocaine, and a 6F introducer sheath was inserted into the common femoral artery via a Seldinger puncture under ultrasound guidance. Then, a 5F catheter (C2 Cobra; Cook, Inc., Bloomington, IN, USA) was inserted through the sheath to select the iliac artery. After injecting a 50% diluted contrast agent, a contrast fluoroscopy image was obtained. The exact location and diameter of the internal iliac artery were confirmed on the contrast fluoroscopy image, and the size of the balloon catheter (Mustang; Boston Scientific, Galway, Ireland, or Admiral; Medtronic, Santa Rosa, CA, USA) was chosen accordingly. The balloon catheter was inserted into the internal iliac artery along the guide wire without touching the common iliac artery. Balloon catheters inserted into both internal iliac arteries were inflated after delivery and umbilical cord clamping before removing the placenta. After removing the placenta, bleeding foci were located on the uterine wall and controlled, after which the balloons were deflated.

After the cesarean section and incision closure, a 50% diluted contrast agent was injected for pelvic angiography to check the state of the uterine artery and iliac artery (Figure 3). If satisfactory hemostasis of the uterus was difficult to achieve, despite repeated attempts to repair the uterine myometrium, the obstetrician requested uterine artery embolization (UAE), which was performed immediately in the hybrid OR after closing the cesarean section incision.

If a hybrid OR was not available for patients in need of an emergency cesarean section, the procedures were performed in two steps. First, balloon catheters were placed in the bilateral internal iliac arteries, and the sheath and balloon catheter were firmly fixed using a tape cover in a conventional interventional radiology (IR) room. Then, the patient was carefully moved to a conventional OR. Balloon inflation and deflation were performed in the conventional OR before and after placental removal, respectively, and the patient was again moved to the intervention room after the cesarean section to perform pelvic angiography and UAE as necessary.

### 2.3. Assessment

Patients’ demographics, number of cesarean sections and abortion, type of PP, risk of massive transfusion, operation time, estimated blood loss (EBL), amount of intra- and postoperative blood transfusion, perioperative hemoglobin level, hospital stay after operation, balloon time, fluoroscopy time, radiation dose, rate of UAE and hysterectomy and complication-related TBOIIA were assessed.

EBL was calculated based on the volume of the suction container, weight of swabs used during surgery and visual estimation of vaginal blood loss. Fluoroscopy time and patients’ radiation dose were measured until the balloon catheters were placed into the bilateral internal iliac arteries before the baby was out. For radiation dose, DAP and the cumulative air kerma (automatically calculated by the intervention equipment) were used.

## 3. Results

Patients’ demographics and obstetric characteristics are demonstrated in Table 1. A total of 62 patients underwent TBOIIA in a hybrid OR, and 19 patients underwent TBOIIA in a conventional IR room. A planned cesarean section was performed at a median GA of 35.5 weeks (30–38 weeks). Of these 81 patients, 28 had a history of cesarean section, and 8 had an in vitro fertilization and embryo transfer (IVF-ET). Eight underwent a laparoscopic operation, and seven had an abortion. Two had methotrexate (MTX) therapy for an ectopic pregnancy.

Of 81 patients, 77 had PP, and 4 had a low-lying placenta.

Seventy-eight of these patients had a uterus-conservative cesarean section, while three had a planned cesarean hysterectomy. The mean operation time was 116.5 min, and the mean EBL was 1440 mL. The mean postoperative length of stay was 5 days.

Among 62 patients who underwent TBOIIA in a hybrid OR, the mean operation time was 122 min, and the mean EBL was 1290 mL. Nine of these patients (14.5%) received a blood transfusion. The mean hemoglobin levels before surgery, immediately after surgery and within 1 week after surgery were 11.3 g/dL, 10.4 g/dL and 9.2 g/dL, respectively. The mean balloon time was 18.3 min, and the mean fluoroscopy time was 3.5 min. Regarding the radiation dose, the mean DAP and cumulative air kerma were 0.017 Gy/cm^2^ and 0.023 Gy, respectively. Ten out of sixty-two patients (16.1%) underwent UAE postoperatively in the hybrid OR (Table 2).

Among 19 patients who underwent balloon catheter placement in bilateral iliac arteries in a conventional IR room and had an emergent cesarean section in a conventional OR, the mean operation time was 99 min, and the mean EBL was 1947 mL. Six out of nineteen patients (31.6%) received a blood transfusion. One out of nineteen patients (5.2%) returned to the intervention room after surgery for UAE (Table 2).

One out of sixty-two patients who underwent TBOIIA in a hybrid OR and two out of nineteen patients who underwent TBOIIA in a conventional IR room had been diagnosed with placenta percreta with bladder invasion based on preoperative ultrasound and thus underwent cesarean hysterectomy. The patient who underwent cesarean hysterectomy in a hybrid OR underwent TBOIIA and UAE prior to the hysterectomy. Her mean EBL was 4000 mL. The mean EBL of patients who underwent TBOIIA in a conventional IR room was 8000 mL and 5000 mL.

None of the 82 patients developed TBOIIA-related complications, such as arterial injury, thromboembolism and arteriovenous fistula.

## 4. Discussion

With the rising cesarean section rate in recent years, placental abnormalities, including PP and PAS, have become a major obstetric complication [14]. However, not only cesarean section itself but also histories of operative hysteroscopy, suction curettage, surgical termination, endometrial ablation and any treatment or diseases that may affect the integrity of the uterine wall may elevate the risk of placental abnormality [15,16]. IVF-ET can also increase the risk of placental abnormality 4-fold to 13-fold [15].

Although PP does not always cause massive hemorrhage [17], a majority of patients with PP are at an increased risk of massive hemorrhage [2], and PAS is reported as one of the most important risk factors [18]. Thus, antenatal assessment of the risk of massive hemorrhage associated with PP, which is defined as blood loss over 1500 mL or over four transfusion units, is necessary to guide proper patient management to improve maternal and fetal outcomes [2,4,19]. Rac et al. reported a placenta accreta index for patients with PP, which encompasses a combination of the smallest myometrial thickness, lacunae, bridging vessels, number of cesarean sections and placental location [20]. Kang et al. introduced a prediction model for massive transfusion of ≥5 units of packed red blood cells in PP during cesarean section that considered the following elements: (1) maternal age, (2) degree of PP, (3) grade of lacunae, (4) presence of a hypoechoic layer between the myometrium and the placenta and (5) anterior placenta [13]. Liu et al. also provided a prediction model for cesarean hysterectomy in PP, which included ultrasonographic findings of vascular lacunae, central placenta previa and loss of the normal hypoechoic retroplacental zone [21]. According to these PP scoring systems, the higher the score, the greater the probability of morbidly adherent placenta, massive hemorrhage and cesarean hysterectomy.

Various conservative management approaches have been employed to preserve the uterus of women who wish to avoid hysterectomy for reasons such as maintaining fertility [14]. Some of the most common conservative management approaches include an extirpative technique (manual removal of the placenta), leaving the placenta in situ, a one-step conservative surgery (partial myometrial resection followed by immediate uterine reconstruction) and the Triple-P procedure (suturing around the accrete area after resection) [22]. Achieving effective hemostasis intraoperatively and postoperatively is a key factor to consider in uterus-conservative management [23]. To this end, several modalities have been introduced, including uterine devascularization, compressive suture and intrauterine balloon, and studies have reported cases of successful uterine preservation surgery using these modalities in patients with placental abnormality [7,24]. Uterine devascularization is important, not only for reducing blood loss, but also for reducing the risk of intrapartum and postpartum hemorrhage by substantially improving the surgical field until proper hemostasis is achieved [25]. Burchell RC et al. reported that internal iliac artery ligation reduces the pelvic blood flow by 49% and pulse pressure by 85%, thereby facilitating terminal branch clotting [26]. Since Dubois et al. first applied balloon occlusion and embolization of the internal iliac arteries in a patient with placenta percreta [8], a variety of endovascular interventions have been employed for cesarean hysterectomy, as well as uterus-conservative management [27]. Although the roles of endovascular interventions in reducing blood loss in placental abnormality remain controversial, these endovascular interventions have enabled the manual removal of placental tissue [28]. Endovascular interventions can be broadly categorized into three types: (1) perioperative TBOIIA, (2) perioperative prophylactic balloon occlusion of the abdominal aorta (PBOAA) and (3) prophylactic or emergent UAE [27]. The choice of endovascular intervention is dependent on the organization and operator, but these endovascular interventions are reported to be useful for the management of PP with a high risk of massive hemorrhage [7,27,29,30,31].

According to studies that compared the efficacy of TBOIIA and PBOAA, both endovascular interventions are safe and useful for minimizing blood loss in placental abnormality. While no significant differences in the EBL and blood transfusion volume were observed between the two groups, the PBOAA group had a shorter balloon time, fluoroscopy time and operation time, and a lower radiation dose. In contrast, the neonatal APGAR score was better in the TBOIIA group than in the PBOAA group [30,32,33]. In our study, the mean balloon time was 18.3 min, similar to that in the PBOAA group [33]. The mean maternal radiation exposure was higher in the TBOIIA group compared to the PBOAA group [31,33]. However, the risk of fetal deformity as a result of irradiation is extremely low in the third trimester, and radiation exposure below 0.05 Gy does not cause congenital malformation, growth retardation and neurodevelopmental abnormality [34]. Thus, although the TBOIIA group had a higher maternal radiation exposure than the PBOAA group, it is anticipated to have little impact on the fetus. PBOAA requires several rounds of balloon inflation and deflation at a fixed interval throughout the entire operation [30,31,32,33], but TBOIIA requires only one round of inflation after placental removal and, consequently, one round of deflation upon achieving proper uterine hemostasis. Hence, the procedure itself is simpler, in addition to the lack of risk of balloon migration, as balloon manipulation is not required. Further, an 8F to 12F introducer sheath is used for PBOAA [30,31,32,33]; thus, postoperative bleeding control at the puncture site may be more challenging than in TBOIIA, which requires a 6F introducer sheath.

UAE is performed alone or in combination with balloon occlusion catheter placement, and it not only helps reduce intraoperative blood loss but also promotes placental resorption [28,35,36].

One major concern with TBOIIA is how to properly fix the balloon catheter in the internal iliac artery such that it does not migrate during surgery. A two-step procedure, where the patient undergoes intra-arterial balloon catheter placement in a conventional IR room and is then transferred to a conventional OR, is a potent risk factor for balloon catheter migration. There are several reports of adverse events, such as equipment failure, intravascular catheter displacement and cardiorespiratory events, during intrahospital transfer [37,38]. In particular, Meller et al. [12] reported that the incidence of arterial catheter dislocation with a two-step procedure was 12.5%. From this perspective, a hybrid OR, which is a general surgical room equipped with high-quality imaging capability appropriate for IR, is the ideal OR for complex obstetric patients in need of preoperative IR procedures, such as balloon catheter placement [12,35,39]. EBL was higher in the patients who underwent cesarean delivery in a two-step procedure (1947 mL) than in those that underwent both TBOIIA and cesarean delivery in a hybrid OR (1290 mL). Among women who underwent cesarean hysterectomy, EBL in the hybrid OR group was 4000 mL, while EBL in the two-step procedure group was 8000 mL and 5000 mL. As these women had no history of medications or diseases that may cause a coagulation disorder, the possibility of intra-arterial balloon catheter displacement during intrahospital transfer could not be eliminated. Another advantage of a hybrid OR is that emergent embolization could be performed immediately after cesarean delivery without transferring the patient to a conventional IR. As shown here, a hybrid OR can help prevent unexpected complications, such as catheter displacement, and provide a safer treatment environment, as preoperative and postoperative IR management can be performed in a one-step procedure without having the patient moved to a different OR.

Our study provided the benefits of one-step TBOIIA in a hybrid OR to eliminate the potential risk of balloon catheter displacement, thus reducing blood loss during cesarean section and enabling a reduction in radiation dose below 0.05 Gy.

This study has a few limitations. First, we could not compare the group that underwent surgery after TBOIIA and the group that underwent surgery without TBOIIA due to the retrospective nature of this study. However, conducting a randomized controlled trial on women with a history of massive bleeding and those predicted to have massive bleeding based on preoperative imaging study can be ethically problematic, despite the availability of techniques to prevent bleeding. Second, the number of women who underwent a two-step procedure in a conventional IR was smaller than the number of women who underwent a one-step procedure in a hybrid OR; thus, the comparison between the two groups was not statistically reasonable. Finally, ultrasound was the only preoperative imaging study included, underscoring the need for subsequent studies to include preoperative MRI findings.

## 5. Conclusions

In conclusion, while the utility of intra-arterial balloon catheter placement and the specific blood vessel in which the balloon catheter should be placed in women experiencing PP with a high risk of hemorrhage remains controversial, our results show that TBOIIA was effective in reducing intraoperative blood loss, thus ensuring safe placental removal and guaranteeing uterine conservation. Moreover, performing TBOIIA in a hybrid OR eliminates the potential risk of balloon catheter dislocation because the patient does not have to be transferred from a conventional IR to OR, and the radiation dose could be lowered to below 0.05 Gy—a level that does not harm the fetus—owing to the superior imaging capability. Therefore, TBOIIA in a hybrid OR has become an established policy for managing PP with a high risk of hemorrhage at our organization.

## Figures and Tables

**Figure 1 jcm-11-02160-f001:**
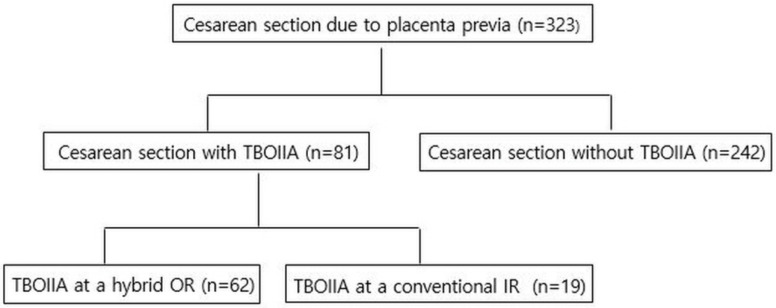
Distribution of patients with placental abnormality who underwent cesarean delivery during the study period. TBOIIA, temporary transcatheter balloon occlusion of bilateral internal iliac arteries; OR, operating room; IR, interventional radiology.

**Figure 2 jcm-11-02160-f002:**
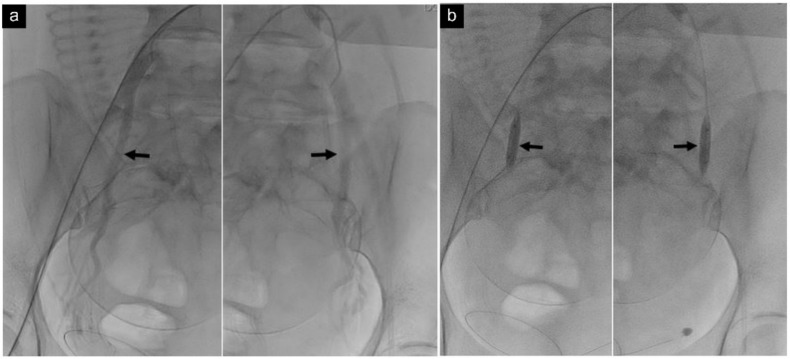
Temporary transcatheter balloon occlusion of bilateral internal iliac arteries. (**a**) Contrast fluoroscopic image of a 40-year-old woman shows the exact level and diameter of bilateral internal iliac arteries (arrows). (**b**) Preoperative fluoroscopic image shows balloon catheters (arrows) in the proximal internal iliac arteries.

**Figure 3 jcm-11-02160-f003:**
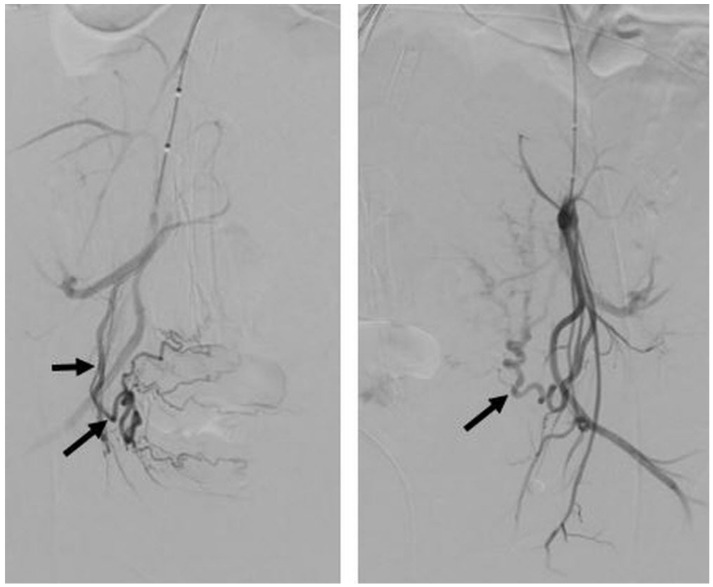
Pelvic angiogram after operation of a 40-year-old woman shows hypertrophied bilateral uterine arteries (arrows) without evidence of contrast extravasation, arteriovenous malformation or abnormal hypervascular lesion. Proper hemostasis was achieved within the operation field, and vital signs were also stable for this patient; hence, the balloon catheter was removed without an additional procedure.

**Table 1 jcm-11-02160-t001:** Demographics and obstetric characteristics of 81 patients who underwent TBOIIA prior to cesarean section. Values are expressed as number (range) or number (percentage). TBOIIA, temporary transcatheter balloon occlusion of bilateral internal iliac arteries; IVF-ET, in vitro fertilization and embryo transfer; MTX, methotrexate.

Characteristic	Value
Age (y)	36.2 (28–45)
Gestational age at cesarean section (wk)	35.5 (30–38)
Maternal history	
Previous cesarean section	28 (34.5%)
History of IVF-ET	8 (9.8%)
Laparoscopic operation	8 (9.8%)
Artificial abortion	7 (8.6%)
MTX treatment for ectopic pregnancy	2 (2.4%)
Type of placenta previa	
Placenta previa	77 (95%)
Low-lying	4 (5%)

**Table 2 jcm-11-02160-t002:** Peripartum outcomes of patients with placental abnormality managed by TBOIIA.

Characteristics	Hybrid OR (*n* = 62)	Conventional IR (*n* = 19)
Type of surgery		
Uterus-conservative CS	61	17
Planned hysterectomy	1	2
Operation time (min)	122	90
Estimated blood loss (mL)	1290	1947
Transfusion rate (packed RBC)	14.5%	31.6%
Hospital stay after surgery (day)	4.8	5.4
Balloon time (min)	18.3	
Fluoroscopy time (min)	3.5	
Radiation dose		
DAP (Gy/cm^2^)	0.017	
Cumulative air kerma (Gy)	0.023	
UAE	10 (16.1%)	1 (5.2%)

CS, cesarean section; IR, interventional radiology room; OR, operating room; RBC, red blood cell; TBOIIA, temporary transcatheter balloon occlusion of bilateral internal iliac arteries; UAE, uterine artery embolization.

## Data Availability

Not applicable.

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
