# Peer review of "The Feasibility and Safety of Temporary Transcatheter Balloon Occlusion of Bilateral Internal Iliac Arteries during Cesarean Section in a Hybrid Operating Room for Placenta Previa with a High Risk of Massive Hemorrhage"

_jcm, 2022, doi:10.3390/jcm11082160_

Round 1

Reviewer 1 Report

the article is interesting and novel, please improve english literature

Author Response

Response to Reviewer 1 Comments

Comments and Suggestions for Authors;

The article is interesting and novel, please improve English literature

Response: Thank you very much for your opinion and review of our manuscript. We carefully reviewed our manuscript and checked English again.

Reviewer 2 Report

I had reviewed this article entitled” Feasibility and safety of temporary transcatheter balloon occlusion of bilateral internal iliac arteries during cesarean section in hybrid operating room for placenta previa with a high risk of massive hemorrhage”. Up to date, there are limited studies regarding bleeding control for those with PP & PAS. Authors provided new evidence of “TBOIIA” method to control intraoperative bleeding and the data showed to be effective. After reviewing this manuscript, I had some suggestions as following:

  • Line 70-79 and Line 80-90 are same, please revise them.
  • In line 72, how to make the diagnosis of placenta previa? By ultrasound? By which criteria? Posterior placenta or anterior placenta previa?
  • Is there any preoperative tool for the diagnosis of PAS in your study?
  • When to perform the planned cesarean section?
  • In line 84, what is the purpose of this prediction model? Please clarify it.
  • In the TBOIIA procedures, what is the method of anesthesia? Local or general anesthesia?
  • Line 116, what is the definition of unsatisfactory hemostasis?
  • How to monitor the complications of TBOIIA during operation? For examples, the dislocation of the balloon.
  • In the table 1 and line 157-158, the types of PP(marginal and partial) are not used currently according to the FIGO definition. Please revise the table. Also, the risk of massive transfusion by the prediction model should be described below this table.
  • Line 125-132, the protocol demonstrated postoperative UAE should be arranged for emergent cesarean section in the conventional IR. However, the results showed only one patient received UAE in this group. Why is the difference?
  • I think it is better to add the neonatal information after cesarean section. Please add more information regarding the neonatal healthy status and complications.
  • Line 283-293, please add the “strength” of your study.
  • In the conclusion, the results reported TBOIIA is effective to control the blood loss for those with PP, not reported the comparison between TBOIIA and TBOAA, so line 297-298 should be revised. Besides, authors did not compare the dislocation rate between hybrid OR and conventional OR, so line 299-302 should be revised.

Author Response

Response to Reviewer 2 Comments

Comments and Suggestions for Authors;

I had reviewed this article entitled” Feasibility and safety of temporary transcatheter balloon occlusion of bilateral internal iliac arteries during cesarean section in hybrid operating room for placenta previa with a high risk of massive hemorrhage”. Up to date, there are limited studies regarding bleeding control for those with PP & PAS. Authors provided new evidence of “TBOIIA” method to control intraoperative bleeding and the data showed to be effective. After reviewing this manuscript, I had some suggestions as following:

  • Line 70-79 and Line 80-90 are same, please revise them.

Response : First of all, we are very pleased to hear your opinion about out manuscript and appreciate your kind review and comments.

As you mentioned, those two paragraphs described similar contents about “Patients”. We removed the second paragraph (line 80-90).

  • In line 72, how to make the diagnosis of placenta previa? By ultrasound? By which criteria? Posterior placenta or anterior placenta previa?

Response : All patients included in the current study took ultrasonography regularly during pregnancy, then they were diagnosed with placenta previa, usually during the second and early third trimester. According to their medical records and ultrasonographic findings, the type of placenta previa was categorized into 4 types; 1) complete type, where the placenta completely covered the internal os; 2) partial type, where the placenta partially covered the internal os; 3) marginal type, which just reached the internal os, but did not cover it; 4) low lying type, which extended into the lower uterine segment but did not reach the internal os. The location of uterus was also confirmed by ultrasonography. As you asked below, we reestablished the classification of types of placenta previa.

  • Is there any preoperative tool for the diagnosis of PAS in your study?

Response : All patients included in the current study were diagnosed with placenta previa by ultrasonography, and the assessment of the risk of a massive hemorrhage and PAS were done by ultrasonography. Recently, when obstetrician found a clues for the risk of massive hemorrhage or PAS during routine ultrasonography, they recommended patients taking MRI. MRI findings have shown promising results for confirming PAS. Therefore, further study will be needed in the future.

  • When to perform the planned cesarean section?

Response : Mean gestational age at cesarean section in the current study was 3.5.5 weeks (30-38 weeks). Originally, obstetrician planned cesarean section at 35 to 36 weeks. However, if patients needed to have emergency cesarean section, they performed surgery before 35 weeks. In case of patients who were referred from other institute, cesarean section was performed at gestational age of 38 weeks.

  • In line 84, what is the purpose of this prediction model? Please clarify it.

Response : As you mentioned at the first review, “ Line 70-79 and Line 80-90 are same, please revise them”, we removed the second paragraph (line 80-90). Therefore, there was no need to clarify the prediction model.

  • In the TBOIIA procedures, what is the method of anesthesia? Local or general anesthesia?

Response : TBOIIA was performed under local anesthesia with lidocaine. We described it at “2.2. TBOIIA procedures”.

  • Line 116, what is the definition of unsatisfactory hemostasis?

Response : The definition of ‘unsatisfactory hemostasis’ depended on obstetrician’s decision. When they thought it was going to be difficult to control bleeding, despite repeated try of repair of the myometrium of uterus during TBOIIA, they asked the interventional radiologist to perform UAE.

  • How to monitor the complications of TBOIIA during operation? For examples, the dislocation of the balloon.

Response : After the cesarean section finished, the pelvic angiography was performed in a hybrid OR to check balloons, the state of arteries, and any types of complications related to TBOIIA. We also checked patients’ symptoms and signs, including arterial pulse of the lower extremity, skin color changes, or etc. The dislocation of balloon was not the complication of TBOIIA in a hybrid OR, because received surgery in the same table where they undertook TBOIIA. They did not have to be moved to other room. Therefore, there was no chance balloons to be dislocated.

  • In the table 1 and line 157-158, the types of PP(marginal and partial) are not used currently according to the FIGO definition. Please revise the table. Also, the risk of massive transfusion by the prediction model should be described below this table.

Response : We have revised the classification of placenta previa; 1) placenta previa, where the placenta is covering cervical os; 2) low-lying placenta, where the placenta edge is within 2cm of the cervical os. We have modified the table, too.

As you mentioned before, the second paragraph of “Patient; line 80-90” has been removed. Thus, it seems to be no longer needed to put “Risk of massive transfusion” to Table. The reason why these 81 patients were included was explained in “Patients” section, and the reference was provided.

  • Line 125-132, the protocol demonstrated postoperative UAE should be arranged for emergent cesarean section in the conventional IR. However, the results showed only one patient received UAE in this group. Why is the difference?

Response : “Line 125-132” explained the two steps procedure of TBOIIA, which the balloon catheters were placed in the conventional interventional radiology (IR) room, then patients were moved to a conventional OR for cesarean section. If a hybrid OR was not available, obstetrician decided to take two steps procedure of TBOIIA in the conventional IR room.

In case of patients who took two steps procedure of TBOIIA, if they needed angiography and UAE, they should be moved to the conventional IR room. In the current study, one of 19 patients took two steps procedure of TBOIIA returned to the conventional IR room after surgery for UAE. Whereas, ten out of 62 patients who took TBOIIA in the hybrid OR underwent UAE postoperatively in the hybrid OR.

  • I think it is better to add the neonatal information after cesarean section. Please add more information regarding the neonatal healthy status and complications.

Response : Our study started from the concept that balloon occlusion of bilateral internal iliac arteries can help reduce blood loss in patients with placenta previa who are expected to have a high risk of massive hemorrhage during cesarean section. Therefore, institutional review board (IRB) approval was obtained for patients with placenta previa. We did not receive IRB approval for their children. Thus, providing neonatal information in the current study is not possible, because it may bring an ethical issue.

However, we believe that further study about neonates will be needed in the future.

  • Line 283-293, please add the “strength” of your study.

Response : We have added the strength of the current study at discussion.

  • In the conclusion, the results reported TBOIIA is effective to control the blood loss for those with PP, not reported the comparison between TBOIIA and TBOAA, so line 297-298 should be revised. Besides, authors did not compare the dislocation rate between hybrid OR and conventional OR, so line 299-302 should be revised.

Response : We have revised conclusion.

Technically, the current study did not perform PBOAA. We compared our results of TBOIIA in the hybrid OR to those of PBOAA in published literatures. We agreed that this sentence might provide wrong information about our study. We have revised it.

In case of patients who took two steps procedure of TBOIIA, balloon catheters were removed after the cesarean section in the conventional OR. Therefore, we couldn’t check the possibility of catheter dislocation directly. According to the result of patients who underwent cesarean hysterectomy, EBL in the hybrid OR group was 4,000 mL, while that in the two-step procedure group was 8,000 mL and 5,000 mL, respectively. Thus, we cautiously doubted the possibility of balloon catheter displacement during intra-hospital transfer. We have revised it, too.

Round 2

Reviewer 2 Report

Overall, these revisions are acceptable but still some minor revisions. Please add the below information into article:

  1. The timing of planned C/S
  2. the definition of unsatisfactory hemostasis

Please highlight them in yellow after revision.

Author Response

Manuscript ID: jcm-1652004

Title: Feasibility and safety of temporary transcatheter balloon occlusion of bilateral internal iliac arteries during cesarean section in hybrid operating room for placenta previa with a high risk of massive hemorrhage

Point-by-Point Response

Please note that the changes made do not influence the content, conclusions, or framework of the paper. We have not listed below all minor changes made; however, these are indicated in the revised manuscript.

Response to Reviewer 2 Comments

Overall, these revisions are acceptable but still some minor revisions. Please add the below information into article:

  1. The timing of planned C/S
  2. the definition of unsatisfactory hemostasis

Please highlight them in yellow after revision.

Response: We are very pleased to hear your opinion about out manuscript and appreciate your thoughtful suggestions and insights.

We have revised our manuscript and added information into our manuscript.

This manuscript is a resubmission of an earlier submission. The following is a list of the peer review reports and author responses from that submission.

Round 1

Reviewer 1 Report

The authors present their study which aims to determine the efficacy and safety of temporary transcatheter occlusion of bilateral hypogastric arteries in patients with placenta accreta spectrum. This is a well designed study and well written manuscript.

Reviewer 2 Report

The authors of this retrospective study investigated the feasibility and safety of temporary transcatheter balloon occlusion of bilateral internal iliac arteries (TBOIIA) during cesarean section in hybrid operating room for patients with placental abnormalities. The results of the experimental group were compared to 19 similar women who underwent TBOIIA in conventional IR room without statistical analysis. The topic is of interest; however, this manuscript has some issues:

- To emphasize the usefulness of TOBIIA, I think it is important to compare it with the group that did not perform TOBIIA. Why didn't you do a statistical analysis comparing the group with TOBIIA (n=81) and the group without TOBIIA (n=242) with medical record review?

- Information on balloon time, fluoroscopy time, and radiation dose of the conventional IR group is also necessary. Although the number of conventional IR groups is small, why not perform a statistical comparison with the hybrid OR group? Because there are more placenta percreta than hydrid OR in 2 patients in the conventional IR group, the estimated blood loss can naturally be higher in the conventional IR group.

- Does radiation dose mean the dose area product (DAP)? The unit of DAP is expressed as µGym2 or Gycm2. If the radiation dose is not DAP, a detailed explanation of how it was measured is required. It is also recommended to attach the cumulative air kerma if possible.

- Three grades of abnormal placental attachment have been defined according to the depth of attachment and invasion into the myometrium: accreta (chorionic villi attach to the myometrium), increta (chorionic villi invade the myometrium), and percreta (chorionic villi penetrate the uterine serosa). The risk of bleeding increases according to the degree of myometirum invasion. Have you not made a distinction about this?

- The transfusion rate is too low compared to other studies. Does the transfusion volume include both surgery and hospital stay?

- Do you have any data on the amount of transfusion or massive transfusion as well as simply receiving a transfusion?

Minor comments

In Abstract 17, EBL should be written in full name.

The angiogram in Figure 3 seems to have been performed with a balloon catheter, but this is not recommended.

Reviewer 3 Report

  1. Summary for jcm-1581630: Feasibility and safety of temporary trans catheter balloon occlusion of bilateral internal iliac arteries during cesarean section in HYBRID operating room for patients with placental abnormalities. This is a retrospective cohort study of 62 women who underwent placement of an intra-arterial occlusive device. The device was placed in the same OR where the surgical procedure was performed. All 62 women had a diagnosis of placenta previa and 1 of the 62 had a diagnosis of PAS. 34.5% of the women had a prior CD. The authors report mean EBL and need for additional surgery or transfusion. Only the woman with PAS required hysterectomy and transfusion. The authors conclude that the device placement is safe when patients undergo surgery for this condition.

  2. The strengths of the manuscript are that the reported use of an intra-arterial occlusive device among women with placenta previa but without PAS in an OR equipped for IR.

The perceived limitations are that there is no evidence that surgical outcomes among women with placenta previa are improved bby placement of the occlusive device.

  1. In its current form, this manuscript does not add to the literature regarding management of PAS.

  2. Title
    The title is somewhat misleading as the majority of women has placenta previa or low-lying placenta without PAS.

Abstract or Summary:
The abstract and methods clearly state that the patient population are 62 women who underwent device placement by IR in a specifically equipped and designated OR prior to CD for placenta previa. However, in the results, the authors refer to an additional 19 women who underwent a similar procedure but in a traditional OR after having the occlusive device placed in the IR suite. The authors should specify that their pain population is women with a prenatal diagnosis of placenta previa and not of PAS.

Introduction
The introduction also discusses the management of PAS but the population is that of placenta previa where the majority of women had no history of CD.

Materials and Methods
In the study purpose, the authors state that the will evaluate the safety of the intra-arterial occlusion of the internal iliac arteries during cesarean section in a specifically designated OR among patients with placental abnormalities. It is not clear whether the objective is to demonstrate the benefit of the designated OR, the balloon occlusion or for the placental abnormalities. This term is very vague and the specific type of abnormality being evaluated should be specified.

Reference 14 is not relevant to this section. This reference is not appropriate, since the variables used to determine risk for massive transfusion included those associated with PAS.

The study design is that of a retrospective cohort study. What is lacking in definition is the population being tested. The intervention and outcomes are not clearly defined. The authors should specify what they mean by “massive blood transfusion during cesarean”.

The results included patients who underwent balloon occlusion in a designated radiology suite; they were emergency cases and really are not comparable to the main study group. The authors also discussed three cases with a diagnosis of PAS, which are not the same as the total study group. The results do not follow the plan and order that was stated in the methods section.

Table 1 is missing information regarding the total number of women included to create the table.

Figures 1-2 and 3 are very good.

The discussion and its take home message should not include references to PAS, as only three women with confirmation of this disorder were included.

The conclusions are not supported by the data presented.